# Train Classification Using a Weigh-in-Motion System and Associated Algorithms to Determine Fatigue Loads

**DOI:** 10.3390/s22051772

**Published:** 2022-02-24

**Authors:** Mariia Zakharenko, Gunnstein T. Frøseth, Anders Rönnquist

**Affiliations:** Department of Structural Engineering, Norwegian University of Science and Technology (NTNU), 7491 Trondheim, Norway; gunnstein.t.froseth@ntnu.no (G.T.F.); anders.ronnquist@ntnu.no (A.R.)

**Keywords:** train classification, static load, wheel load, vertical force, measurement system, weigh-in-motion (WIM)

## Abstract

This paper presents a methodology for classifying train passages into different types with a weigh-in-motion (WIM) system to allow the calibration of railway fatigue load models and identify individual vehicles from the measurements for the continuous calibration of railway WIM stations from in-service trains. The quality assurance of the measured responses is demonstrated using statistical methods. This paper discusses the measurement station, the method used for processing the raw data, the algorithm used to identify the train types and vehicles automatically, and the limits of the obtained load spectra. The measurement errors are demonstrated to be satisfying for use in fatigue load model calibration. Furthermore, this paper proposes actions for accurately obtaining the actual traffic conditions and describes the future work required in this area.

## 1. Introduction

Railway wheel loads’ monitoring data are of key importance in monitoring the dynamic behavior of a vehicle and railway track, and in the online tracking of actual train loads and possible imbalances. However, they are also used in the study of traffic safety and in the maintenance planning of the track and bridges, which are very important parts of the railway system.

The wheel loads can be obtained by directly measuring the wheel rail contact force [1] or indirectly detecting the force applied to the infrastructure. The monitoring of traffic loads from the infrastructure viewpoint allows a train set to be measured using one measurement system which can be installed on the bridge (bridge weigh-in-motion system, B-WIM [2,3]) or on the track (weigh-in-motion system, WIM [4,5]). The WILD system (wheel impact load detector [6]) is based on the same principle as the WIM system and is used for the identification of damaged wheels.

The B-WIM system [2] measures deformation on the level of bridge and calculates axle weights using Moses algorithm [7]. The basic of this algorithm is to minimize the difference between the measured and predicted bridge response based on the Influence Lines (IL) concept [8]. However, compared to WIM system, the B-WIM system is less sensitive to the dynamic effects and highly depends on the bridge type.

The purposes and the methods of data processing in WIM stations depend on the intended implementation of the measurement system. For instance, in study [9], the system is installed on sleepers and the purpose of the described algorithm is to accurately estimate wheel loads as well as imbalances at high vehicle speeds. The authors of [10] identify the weight of a moving train with the help of the minimization of the time history of strains at the rail’s foot and its difference from its analytical counterpart. The authors of [11] develop the wavenumber domain method (WDM) with the goal of identifying the wheel–rail force waveform through the monitoring of the rail dynamic response. In study [12], the simultaneous structural health monitoring of the railway track and train wheel is performed using data from fiber-optic sensors and dedicated algorithms.

However, the traffic conditions are efficiently obtained by measuring the axle loads and pitch while the train passes a measurement station, WIM system. In study [13], authors discuss the factors affecting the accuracy of WIM systems and the calibration of such systems. Traditional WIM systems measure strains in rails using electrical strain gauges, which are the most popular classical sensor type [4], fiber-optic sensors (distributed [14] and local [5]), or piezoelectric sensors [15]. All these systems demonstrate a good correlation between the responses of nonclassical sensors and those of strain sensors. The listed articles focus on the weighing of single trains using different techniques; however, their methodologies can be used for the estimation of the remaining fatigue life of railway bridges. 

Fatigue is one of the primary damage mechanisms of steel railway bridges [16] due to the higher loads involved and the intensity of current traffic [17]. The description of the loading cycles experienced by the material is necessary in order for both approaches to estimate the remaining fatigue life: a combination of a fatigue endurance model [18] and a damage accumulation model is used [19], or a fracture mechanics approach [20]. The fatigue load model is a representation of the traffic conditions at a bridge. A Norwegian load model based on rolling stock data, operational rules, and statistics has recently been presented [21]. The Norwegian project, which involves fatigue life estimation [22,23], showed that 7 out of 21 studied steel railway bridges in the Norwegian railway network had exceeded their remaining fatigue life long ago, although the inspection of these bridges had not shown any sign of fatigue cracking. The remaining fatigue life of railway bridges can be more accurately estimated by introducing calibration factors for the load model from measurements of current traffic conditions.

The classification of train class (type) is crucial for the calibration of load models because the response from a passenger train is significantly different from that of a freight train or a local suburban train. However, there is no simple and efficient methodology for classifying train types from WIM measurements. This paper presents an algorithm for the automatic identification of train types as well as locomotives and wagons based on a priori information to fill this gap. The method to process raw WIM data with the precision sufficient for fatigue load model calibration is also discussed. The output of the algorithms is a database of trains which crossed the station with features important for fatigue load model calibration: train and locomotive type, load distribution, geometry, direction of moving, speed, and temperature at the site as additional information. The identification of vehicles with known geometries and loads is also valuable because it provides the means to continuously calibrate measurement stations from in-service trains.

This paper is organized as follows. First, the methodology used for obtaining the traffic conditions is presented. Next, the data processing method and proposed algorithm for the identification of vehicles and train types is shown. Next, the uncertainty of the obtained loading spectra is discussed. Finally, several conclusions regarding the methods and results are drawn, and future perspectives and plans are presented.

## 2. Materials and Methods

### 2.1. Case Study and Monitoring System

The case study considered in this work is a measurement station located on the Sokna bridge by Lundamo station on Dovrebanen (Dovreline), which connects Oslo and Trondheim. The location of the measurement station was chosen primarily due to the low permissible speed at the site (50 km/h). At lower speeds, the dynamic part of the loading is reduced, which enables the better determination of the static part of the loading. Furthermore, the sensors of the system were placed on the bridge to reduce the bias in the measurement due to the stiffness variation in the substructure of the tracks and to obtain the exact speed of the train during the passage of the bridge.

The monitoring system was installed and put into operation in January 2021; it is still collecting data at the site. The current total number of trains measured in this research is 8224; these measurements were collected over 7 months. It has been observed that approximately 50 trains per day pass over the measurement station on weekdays, while 20 trains pass per day on weekends. The majority of the traffic is local suburban passenger trains (multiple units). Although freight trains account for approximately 25% of the traffic, freight traffic should be monitored carefully because the damage potential of freight trains is much higher than that of passenger trains [17]. Information about train types is presented in Section 2.5, and the precise characterization of traffic is shown in Section 3.1.

The monitoring system used consists of sensors, a datalogger, a network, and a remote server for storage, data processing, and data distribution. Two types of sensors are used in the system: strain and temperature sensors. Figure 1 gives an overview of the nominal placement of the sensors in the system and a perspective view of the site.

### 2.2. Temperature Measurements

The monitoring system was equipped with a temperature sensor (PT 100 Sensor according to EN 60,751—DIN 43,760, manufacturer is Roth+Co., Oberuzwil, Switzerland). The data were logged in ohms and then converted to Celsius using the Callendar–Van Dusen Equation (1) with parameters *A* = 3.9083 × 10^−^³ *B* = −5.775 × 10^−7^, and *C* = −4.183 × 10^−12^. Parameters A, B, and C are usually obtained experimentally from measurements of resistance at temperatures of 0 °C, 100 °C, and 260 °C, respectively.
(1){Rt=R0 (1+At+Bt2),  t ≥ 0 °C Rt=R0(1+At+Bt2+C (t−100)t3),  t<0 °C
where t is the temperature in °C, $R_{t}$ is the resistance at temperature t, and $R_{0}\;$= 100 ohms is the resistance at 0 °C.

The range of measured temperatures (Figure 2) was approximately 50 degrees, and this provided us with an opportunity to study the correlation between the monitored strains and temperature. The results of the correlation study are presented in Section 3.2.

### 2.3. Strain Measurements

The monitoring system included six channels of two 350 ohm chevron 6 mm strain gauges from HBM. The specification type was CXY41-6/350HE and the sampling rate was 5000 Hz. The strain gauges were connected in full-bridge configurations glued to each rail on the neutral axis. The longitudinal distance between each channel was nominally 3 m, as this distance ensured accurate speed estimation even at low sampling rates (less than 10% error at 200 km/h and 100 Hz). Figure 3 presents the placement of the strain gauges of one channel on the rail cross section and the nominal placement relative to the sleepers.

The strain channels (sensors) were designated L1–3 and R1–3, with L/R referring to the left and right rails facing Trondheim and 1–3 referring to the position along the track. The data measured in strain channels were used to calculate wheel loads with the help of Equation (2) [24]:(2)$$F\left(t\right)=V\cdot \frac{GF}{4}\cdot \left[\varepsilon_{1}\left(t\right)-\varepsilon_{2}\left(t\right)+\varepsilon_{3}\left(t\right)-\varepsilon_{4}\left(t\right)\right]\cdot CF$$
where F(t) is the external vertical force from the wheel to which the rail section between the strain gauge pairs is subjected, $\varepsilon_{n}\left(t\right)$ is the output according to the numeration of the measuring grids in Figure 3, *GF* is a gauge factor, *V* is the input excitation voltage, and *CF* is a calibration factor that converts the voltage output of the system to tonnes. The output of the measurement system (signal) is the voltage ratio placed in brackets: $\left[\varepsilon_{1}\left(t\right)-\varepsilon_{2}\left(t\right)+\varepsilon_{3}\left(t\right)-\varepsilon_{4}\left(t\right)\right]$. A general procedure used to obtain a calibration coefficient $cc=V\cdot \frac{GF}{4}\cdot CF$ from the ratio of voltage to tonnes is presented in Section 2.6.

### 2.4. Data Processing

An example of the output signal of one strain channel of the measurement station is presented in Figure 4a. According to study [14], the negative peaks in the strain signals represent the tension that the rail experiences just before the wheel passes the gauge, while the positive local maximums represent the compression of the rail after the wheel passes the gauge. Therefore, each high-strain peak value (spike) corresponded to a wheel of the passing train. To describe the stress histories, the load was defined by the static load function f(x) of the train along a spatial coordinate x (Figure 4b). 

Data logging was triggered by the exceedance of a threshold value at one of the sensors (L2) and data were stored 5 s before and after the threshold is exceeded. Raw strain and temperature data, together with the triggering date and time, were stored in a file locally and copied to the remote server every hour.

The raw data were processed in the following way to extract relevant information about the train for load model calibration and quality assurance.

Each wheel passage was identified by peak detection on the signal. The peaks and corresponding indices were used together with information about the measurement system (e.g., sensor position and sampling rate) to determine the wheel load, pitch (distance between wheels), and speed. Train properties, such as the train type, kind of locomotive being used, number of axles, and running direction, could subsequently be determined by the synthesis of wheel information and other (a priori) information. The formats of the extracted wheel and train information are given in Table 1. The subsections below give a more detailed description of the approach used to extract the information about wheels and trains. The train and locomotive classification was performed at this stage, but the details are presented separately in Section 2.5 due to the topic’s importance.

#### 2.4.1. WheelVoltageRatio

The wheel voltage ratios *(WheelVoltageRatio)* were obtained with the following procedure. First, the raw signal from the channel was filtered using a moving average filter to remove random noise and smooth the signal. Then, the final signal was obtained by removing the bias in the signal (Figure 5a). The peak value was defined as the value of the filtered signal corresponding to the middle of the distance between up- and down-crossing a threshold (2.5 × 10^−5^); see Figure 5b. The signals from two adjacent sensors detecting the same bogie are presented in Figure 5c.

Wheel voltage ratios were transferred to wheel loads in tonnes through *cc*; see Equation (2). The coefficients of transferring signals from channels to the train axle loads (calibration coefficients) were obtained through the comparison of the measured strain data with the load axles of known trains. Details of this procedure are given in Section 2.6.

#### 2.4.2. WheelSpeeds, TrainSpeed and TrainDirection

The speed of the detected wheel (*WheelSpeeds)* was calculated as the distance between sensors divided by the time between the detection of the peak at one sensor and the detection at the second sensor, see Equation (3). The time between detections was a product of the number of samples between peak arisings and the sample interval.
(3)$$v=\frac{dx}{\left(t_{2}-t_{1}\right)dt}$$
where v is the speed of the detected wheel, dx is the spatial distance between sensors from which the signals were obtained; dt is the sampling interval of the measurement system—i.e., the time between each sample, and $t_{2}$ and $t_{1}$ are the number of a sample of the peak detection at the second and first gauges, respectively. The sign of the speed defines the direction of the train *(TrainDirection)*. The train speed (*TrainSpeed)* is defined as the mean value of wheel speeds in the train; see the distribution of train speeds in Figure 6.

#### 2.4.3. WheelPitch

The wheel pitch *(WheelPitch)* is the distance between two adjacent wheels and the most important information in train type classification. This was calculated as the number of samples between the load peaks detected at the first gauge multiplied by the distance the train travels between samples; see Equation (4). This distance was calculated as the ratio of the distance between two gauges dx and the mean value of samples between peaks detected at the first gauge and at the second gauge.
(4)$$pitch=dt_{1}\frac{dx}{mean\left(t_{2}-t_{1}\right)}$$
where $dt_{1}$ is an array of the numbers of samples between the load peaks detected at the first gauge; $t_{2}$ and $t_{1}$ are the arrays of peaks detected at the second and the first gauges, respectively; and $mean\left(t_{2}-t_{1}\right)$ is the arithmetic mean of the difference between these two arrays.

### 2.5. Train and Locomotive Classification (TrainType and LocomotiveType, IsElectric)

The compilation of a train type database is the first stage in train and locomotive classification. There are two main types of trains operating in Norway: passenger and freight trains. Passenger trains can be further categorized as regular passenger trains and multiple units (MUs) with three classes—92, 93, and 73—as shown in Figure 7. The latter type does not have a locomotive and is set in motion by an engine placed in one of the wagons. The number of wagons in such trains is generally much lower than that in trains driven by locomotives. These types of trains have the same geometry from passage to passage and can be identified from a priori information. Note that several MUs can be connected together to form a larger train (for example, Class 93, multiple operation; see Figure 7).

There are three types of wagons in service in the Norwegian railway network [25] (Figure 8): two-axle wagons (2 axles), bogie wagons (4 axles), and Jacobs bogie wagons (6 axles). A bogie wagon contains two or more wheelsets in one framework for better vehicle stability and improved ride quality by absorbing vibrations and minimizing centrifugal impact forces [26]. All vehicles in the study are equipped with two-axle bogies, except for two-axle wagons (no bogie) and type Co’Co’ locomotives (three-axle bogie). According to study [25], the pitch of bogies of freight wagons is always 1.80 m, while for passenger wagons this value is in the range of 2.50–2.70 m.

The method used to obtain the train type *(TrainType)* and locomotive type *(LocoType)* was based on the comparison of the geometry (train pitches) of detected vehicles with the geometry of vehicles from the database adopted from [25]. The comparison of vehicles was conducted using a typical data science approach such as vector analysis. The vector of pitches was defined as an array of vehicle pitches, and every pitch in the vehicle was a coordinate in the space of vehicle pitches. It was possible to compare two vectors in one space. Vectors of vehicles from the database are shown in Table 2 and Table 3.

The Euclidian norm of a vector is defined as the distance of the vector coordinate from the origin of the vector. In particular, the norm may be defined as the square root of the inner product of a vector with itself [27]; see Equation (5):(5)$$norm=\sqrt{\overset{n}{\underset{i=1}{\sum }}a_{i}^{2}}$$
where $a_{i}$ are vector coordinates.

The norm of the difference between vectors, such as database vehicles and detected vehicles, is a measure of the difference between these two vectors. The smaller the norm of the difference is between two vectors, the higher the similarity between them will be. The threshold of the norm for determining if two vehicles are the same based on their geometry is discussed below.

It is important to consider the distribution of norms to study the nature of a variable, make an accurate estimation of the threshold, and distinguish different types of trains and vehicles. The norm, and consequently the threshold, grows with the number of axles. As the first step of approximation, the threshold for Class 93 is 0.6, that for Class 92 is 1.5, and that for Class 73 is 3. Locomotive identification consists of two steps: calculating the norm for each locomotive type and choosing the type with the minimal norm. This is a locomotive type that is the most similar to the measured locomotive. The threshold for locomotive identification is 0.6 for locomotives with 6 axles and 0.25 for locomotives with 4 axles. The database of reference locomotives should be constructed accurately to exclude inaccuracy in identification. The absence of actual traffic locomotives with a geometry similar to that of a locomotive from the database can lead to the first locomotive being defined as a similar locomotive from the list. There is also a limitation regarding the length (sum of pitches) of the supposed locomotive: a locomotive with 4 axles is supposed to be in the range of 8 m to 17 m, while the length of a locomotive with 6 axles is from 9 m to 22 m. The remaining trains are defined as trains with unknown locomotives, and the limit of the norm is 1.3 (locomotives that are not in the database of locomotives, locomotives that are similar to one from the database, but where the norm exceeds the threshold). If the train is defined as MU (class 73, 92, or 93), the locomotive type is none, which indicates the absence of a locomotive.

The algorithm used to identify train types is presented in Figure 9. The first step is the locomotive geometry test, which is carried out to identify the locomotive type or its absence. If any locomotive is identified, then wagon identification is performed. If no wagon is identified and the locomotive is identified, the type of train is a single locomotive. If a two-axle wagon or a bogie pitch measuring less than 2 m is identified, the train is a freight train; otherwise, the train is a passenger train. A train without a locomotive will pass the MU geometry test. If the result is positive, the output of the test is the type of MU; otherwise, the last test will be performed. If the number of axles in the train is less than 6, the train type is a service train; otherwise, the train type will be unidentified. The results of the filter are shown in Section 3.1. Only 3% of all trains are unidentified, and the train group consists of incorrectly measured MUs, trains with incorrectly measured locomotives, and long service trains.

According to Table 4, the type of force setting a train in motion is identified; in short, this answers whether a train is electrical *(IsElectric)*: ‘true’, ‘false’. This feature can be used for studying the influence of electrical current on a monitoring system. However, no correlation between whether the train was electrical and any parameters of the train was detected.

### 2.6. Calibration Coefficient for Channel

After the identification of train types, the calibration coefficients of the strain gauges transferring the voltage ratio to tonnes could be obtained by comparing the measured axle loads of locomotives with their known axle loads (reference loads) due to the constant axle weights of locomotives; see the table of reference loads, Table 5. The axle loads of locomotives in Table 5 were adopted from [25], where the axle loads are rounded to the nearest whole numbers. The wheel loads were derived by division of axle load to the number of wheels in the axle (2), the weights of locomotives were derived by multiplying the axle load to the number of axles of corresponding locomotive (4 for Bo’Bo’ type, 6 for Co’Co’ type). Most locomotives used in this study were electric locomotives that did not contain fuel; thus, the axle weights of the locomotives were not affected by the difference in the amount of fuel.

The axle load is a variable that is used to calculate the damage that the passage of a train causes to the infrastructure. Since gauges are placed on rails, they measure wheel loads. The axle load is the cumulative load of two wheels: right and left. However, it is not permitted to sum signals from the left and right channels even if they are placed against each other. The reason for this is the different calibration coefficients used for different channels. Table 5 presents the wheel load of the ideal case when the load is equally distributed across the left and right wheels. The assumption of a load being equally distributed between the wheels of locomotives is adopted for further research.

The calibration coefficient for one axle *i* for channel *j* is defined as the ratio between the expected axle load in tonnes and the response of the channel. Taking into account the assumption of equal distribution, the calibration coefficient for wheel load is twice as low as the axle calibration coefficient; see Equation (6):(6)$$cc_{i,\;j}=\frac{1}{2}\cdot \frac{p_{exp,\;\;i}}{\varepsilon_{meas,\;\;j}}\;$$
where $cc_{i,\;j}$ is the calibration coefficient for wheel of axle *i* measured by channel *j*, $p_{exp,\;i}$ is the expected axle load for axle *i*, $\varepsilon_{meas,\;j}$ is the response measured in the channel *j* during the passage of the wheel. 

The indicator that the calibration coefficient for the channel was chosen correctly is the most fit mean value of the distribution of calibrated loads and the lowest coefficients of variation for these distributions. Additionally, the measured weight of the locomotive being close to the value in the database is a sign that the measurements were good. To achieve this goal, the method of least squares, a statistical procedure used to obtain the best fit line to data, minimizing the sum of residuals of points from the line using linear algebra and simple calculations [28], was chosen to obtain the calibration coefficients for channels; more information on this is provided in Section 3.2 (measurement error). The correction coefficients were calculated ignoring the measurements of first axles in bogie due to the high dynamic effects that the first axle was subjected to and, therefore, the high standard deviation of the distribution of signals measured for the first axles of bogies.

### 2.7. The Final Value of Axle Load

After determining the calibration coefficients for channels L1-3 and R1-3, a product of the measured signal and calibration coefficient was added to the file containing wheel information for each channel for each axle. The final question was how to proceed with the channel loads.

The question of how to handle damaged wheels is of high importance in the literature [29]. Dynamic effects due to wheel flats or irregularities on the rail surface can produce extreme (outlier) measurements from sensors on either the left or right side. To minimize these outlier effects, the median value of the measured wheel loads was taken as the final wheel load—i.e., the wheel load was defined as the median of the wheel loads estimated by sensors L1-3 and R1-3 on the left and right sides, respectively. The final value of the axle load was calculated as the sum of the mean values between channels from the left and right sides; see Equation (7):(7)$$p_{axle}=median\left(w_{L1},w_{L2},w_{L3}\right)+median\left(w_{R1},w_{R2},w_{R3}\right)$$
where $l_{i}$ is the final load obtained from channel *i*.

The median value of channels of one side ignores outliers in the results of channels of one rail. The sum of values obtained from the left and right sides ensures a comprehensive representation of the axle load. The final step to obtain loads is averaging loads in bogie; see the details in Section 3.3.

## 3. Results and Discussion

### 3.1. Overview of Traffic at Site

The traffic on the site consisted of passenger and freight trains, as shown in Figure 10a. The majority of the trains were MUs (Class 73, 92, and 93). Class 73 is regional passenger traffic between Trondheim and Oslo, while Classes 92 and 93 are local suburban trains around the city of Trondheim. Approximately 55% of all identified trains were local suburban (LS) passenger trains (Class 92, 93), 16% were regional passenger trains (regular passenger and Class 73), and 26% were freight trains. The distribution of locomotive types is presented in Figure 10b. The main locomotive types were Bo’Bo’-f, Bo’Bo’-c, and Co’Co’-b.

### 3.2. Quality Assurance of the Data

The quality assurance of the measured data is important for the fatigue load model calibration which is the reason for measurements. Overestimation of the loads can lead to the underestimation of the remaining fatigue life of the bridge and ineffective spending of finances for the renovation of the structure. Underestimation of loads, in its turn, can lead to carelessness treatment of worn bridges and further collapses of structures.

To conclude, a correlation study was performed for the quality assurance of the monitoring system, and the measurement error was studied. A correlation study aims to define variables that influence the final measurement result. The Pearson correlation coefficient, the relationship of the covariance of two variables, and the product of their standard deviations were used in this section. Measurement error was calculated to investigate the bias of the system. Both studies were performed for each axle separately.

#### 3.2.1. Influence of Speed on Estimated Wheel Loads

The correlation between the speed and signal of the sensors is presented in Table A1 (Appendix A) and Figure 11. Table A1 presents the correlation coefficients, showing the correlation between the speed and signal of sensors in most spread-type locomotives with 4 axles and 6 axles: Bo’Bo’-f and Co’Co’-b. The table includes information on all channels for all axles separately and for all locomotive axles. Figure 11 depicts scatter diagrams of cases with the highest correction coefficients for Bo’Bo’-f (axle 2, R1, L3) and the same cases (axle 2, R1, L3) for Co’Co’-b.

As seen in the table, the correlation is not uniform for all observed cases (composition of the number of axles and the channel). Even the correlation of speed and signal for one channel is not the same for all axles observed. This means that the correction coefficients are not useful for increasing the consistency of the results of the correlation study. The approach presented in this article (summation of the median values of wheel measurements and the further averaging of the loads of bogies) solves this problem. The final results for the axle loads do not correlate with speed.

#### 3.2.2. Influence of Temperature on Estimated Wheel Loads

Due to the wide range of measured temperatures and the possible impact of temperature on the strain gauge measurements, the correlation between temperature and strains was studied. The correlation between temperature and strains is weaker than the correlation between speeds and strains; see Table A2 (Appendix A) and Figure 12. The correlation is not consistent for channels (the value changes between cases), and it is not possible to use a uniform correction coefficient to reduce correlation. However, the final axle loads do not correlate with the temperature.

#### 3.2.3. Correlation between Estimated Pitch from Left and Right Side Sensors

The correlation between the pitch obtained from sensors of the right rail and left rail is 0.999. Figure 13 shows an example of a scatter diagram showing the correlation between these variables using data concerning the distance between the second and third axles of Class 92. This correlation ensures the correctness of the train type identification.

#### 3.2.4. Correlation between Signals from Different Sensors

Table A3 (Appendix A) and Figure 14 illustrate the strong correlation of signals measured in different channels (sensors) using signals measured during the passage of trains of Class 92 going to Trondheim; only the first wheel axle in the bogie is used for plotting. The correlation of strains measured in channels from one rail is stronger than the correlation between channels from opposite rails. A possible reason for this difference is an imbalance in the load distribution between the wheels of one wheelset—for instance, due to accumulating weight on one side of the train or measurement error.

#### 3.2.5. Measurement Error

An assessment of measurement errors was performed using the axle weights of locomotives and geometries of locomotives and MU. The measurement error in the current work is defined as a composition of the type of distribution of measured known axles, its mean value, and the coefficient of variation. The difference between the expected axle weight and the mean value of the measured loads shows how close the measurement is. The coefficient of variation provides an understanding of the quality assurance of the measured value. The closer the mean value of the distribution is to the expected value and the smaller the coefficient of variation is, the smaller the measurement error is considered to be.

An assessment of errors regarding the weights of axles has been proven to be accurate and is used by the Association of American Railroads [30]. In study [31], James also uses such a methodology to assess measurement error. The assessment performed using methodology based on a set of real trains is illustrative because the output of the assessment is a distribution of possible measurement errors compared to measurement strains induced in the infrastructure by the passage of a single train. Moreover, the methodology reduces costs for the validation of the system in comparison to using a specially composed train with known axle loads.

The WILD system used in [31] was installed in 1990. The gauges of this system are welded to the rail. The measurement error is defined by comparing the measured load with the load of the nominal axle load of the locomotive. The author plots a distribution of the second axle of locomotives of one type being measured during half of the year. The coefficient of variation of distribution for the first studied locomotive equals 7%; for the second type of locomotive, it is 4.5%. The mean value of the distribution is higher than the nominal value, which could show the presence of bias in the measurement system. James states that the possible reason for this is the dynamic effects in locomotives. To ensure the absence of dynamic effects, the author studied an empty iron wagon.

Table 6 compares the measurement error of the current measurement station for loads calculated based on wheel loads in each channel individually and for the final result, wheelset load (axle load). Mean values and coefficients of variation are presented for distributions of each axle of measured locomotive type Bo’Bo’-f. The precision up to the first decimal was chosen as enough representative and for preserving clear representation of information as for mean values, as well as for coefficient of variation. The expected value for each axle of locomotive type Bo’Bo’-f is 21 tonnes, and the expected wheel load is 10.5 tonnes. Table 6a shows that the mean value for axle weight in channels varies from 9.4 to 12.1. Table 6b presents the coefficient of variation in the range of 5.6% to 12.6%. The mean values of the axle weights vary from 19.7 to 22.8, with a variation in coefficient of variation from 3.0% to 8.7%. The final results were considered to be sufficiently precise for the purposes of this work. The coefficients of variation were less for the final value than for the values calculated for channels.

Table 7 and Table 8 present the measurement errors for the weights of locomotives and the axles of locomotives, respectively. The correctness of the mean values for axle weights and locomotive weights is higher for locomotives dominating the traffic. Based on previous evidence, conclusions regarding the good quality of the measurement system can be drawn. Moreover, the correlation between axle load (final result) and speed or temperature is low.

However, the common trend discovered is that the weight of the first axle in the bogie is higher than the weight of the second axle. The coefficient of variation is also higher for the first axle. This trend concerning the mean value is noticeable for wheel loads and for final axle weight, but the coefficient of variation is lower for final weight for all cases. According to Table 8, this trend can be observed for all locomotives. A possible explanation for this effect is that the first axle of the bogie is subjected to greater dynamic effects than the second one.

Table A4 and Table A5 (Appendix A) represent the distribution of the relationship between the first and second axles in the bogies of locomotives and wagons for weights from each channel separately and for the final axle load and wheelset load. The first column also contains the number of bogies in the distribution. To represent the very large number of cases in a concise and easy-to-understand manner, each row of the other columns contains three numbers showing the percentage of relationships between axles in bogie where:The first axle is lighter than the second by more than 5%;The first axle is heavier or lighter than the second by no more than 5%;The first axle is heavier than the second axle by more than 5%.

The most common distribution is that most bogies have a trend where the second axle is heavier for locomotives as well as for wagon bogies. However, the main goal of a bogie is to distribute loads [26], and, therefore, measurement error possibly occurs, as mentioned earlier, due to some dynamic effects.

On the one hand, our literature review of this topic has shown that studies using WIM and B-WIM measure different trends, but we have seen no mention of the current trend even if the pictures of train load functions present this trend [5,6,32,33]. The measured vertical displacement of the rail during the passage of the first axle of the bogie is not always larger than the displacement after the passage of the second axle [33,34].

On the other hand, study [2] uses trains with distributed loads between axles in a bogie to verify their methods of measuring wheel loads and estimating real train load distributions of passenger trains with equal axle weights in one bogie. Finally, study [25] presents the geometry of wagons with distributed loads in bogies for Jacobs bogie-type wagons and distributed loads in all axles of two-axle and bogie wagons.

### 3.3. Averaging, Final Axle Loads, and Measurement Error

Based on the arguments presented in subsection “Measurement Error”, we decided to average the loads of the trains. Three approaches for averaging the loads were studied:* average*
*vehicle*, *average*
*bogie*, and *average*
*design.* Each train with a locomotive includes a composition of three types of wagons, such as two-axle wagons (T), bogie wagons (B), and Jacobs bogie wagons (J) (Figure 8). Only the two-axle wagon does not include bogies.

*The average**vehicle* is characterized by an equal distribution between all axles in all types of vehicles (locomotive, T, B, J). *The average bogie* has an equal distribution between the axles of one bogie in bogie vehicles (locomotive, B, J), and the loads of two-axle wagons (T) are not averaged. During *the average*
*design,* the loads of bogie wagons, two-axle wagons, and locomotives (B, T, locomotive) are distributed between all axles of the vehicle, and the loads of Jacobs bogie wagons are distributed between the bogies’ axles. Examples of the load functions of one real measured train (initial) and the averaged functions of the train are presented in (Figure 15). The *average*
*bogie* is considered to be the most appropriate averaging approach based on the analysis of damage caused to the infrastructure by the passage of a train.

To analyze the damage caused to the infrastructure, the influence line (IL) approach [35], a standard procedure used for the prediction of stress histories in the evaluation of existing railway bridges, was used. In IL approach, structural detail is presented as influence line with a certain type and length. IL is the response of a structure subjected to a unit load locating and moving along. The static response is defined by the convolution of the influence line and the load function. The stress ranges are obtained using the rainflow cycle counting algorithm [36]. The fatigue damage caused to the infrastructure is calculated taking into account the endurance curve [37], the dynamic amplification factor [8], and Miner’s damage accumulation rule [19]. A single slope linear fatigue endurance curve (slope parameter b=5) is used in fatigue calculation.

Using the method described, four cases that differed in the load functions of trains, such as the initial load function and the load function obtained using three studied averaging approaches, were studied. The damage was calculated for the most widely spread ILs with lengths ranging from 2 m to 101 m. To compare the damage calculated for four cases, the damage calculated using the initial load function was suggested to be 100% of the damage caused to the infrastructure. The amounts of damage calculated using averaged functions are presented as percentages of the initial damage. Therefore, Figure 16 presents (a) one of the most widely spread ILs and (b) the ratio between the damage calculated with averaging load functions and the damage calculated with the initial load function as a percentage for this IL. The ratios for each averaging approach are compared with the damage caused to the infrastructure calculated for the initial load function, with a red line representing 100% of the damage.

The difference between the damage calculated for the initial load function and the function averaged with *the average*
*vehicle* procedure is the highest because such an approach changes the function the most. The spikes of the load function are equalized, which leads to lower damage being caused in most cases. A common trend in the estimation is that in the short IL cases, the scope of the damage is underestimated. The estimations made using *the average*
*bogie* and *average*
*design* are very similar in their predictions for this train because the vehicles with the highest damage potential, J wagons with 6 axles, are averaged in both approaches in the same way. Although locomotives have a higher damage potential, the number of J wagons is usually much higher in one train, and the averaging of such vehicles has a higher impact on the final damage.

The real cases can differ from the assumed load distributions due to the different placement of passengers and goods in the wagons. In the case of the accumulation of people or goods in one half of the vehicle, the weights of axles under this half will be higher. For this reason, the *Average_vehicle* is supposed to be unrealistic, as well as the averaging of T wagons. On the other hand, as mentioned above, the main goal of the bogie is to distribute loads, and the distance between the axles of one bogie is not so high. Taking into consideration previous reasoning, *Average_bogie* was chosen as a technique for averaging the loads.

The measurement error of this approach (Table 9) is considered suitable for the purposes of the measurements, further fatigue load model calibration, and averaging wheelset loads according to *the average*
*bogie* approach was chosen as the final process of strain data processing. The mean values of axle loads of the most spread locomotives, such as Bo’Bo’-f, Bo’Bo’-c, and Co’Co’-d, differs from the expected loads less than 0.3 tonnes, the coefficients of variation are not higher than 4.5%. Figure 17 shows boxplots representing the distributions of measured loads in all locomotives of (a) the first bogie and (b) second bogie. The top and bottom of each box represent the 75th and 25th percentile values, the whiskers show the minimum and maximum, the horizontal line represents the median, and the triangle represents the mean value.

## 4. Conclusions

This paper presents an approach that can identify train types for fatigue load model calibration and vehicles for the continuous calibration of the railway WIM station. The current method is fully automatized, and the output of the system is information concerning the type, geometry, speed, temperature, and loads of trains crossing the measurement station. The train types were identified using a norm approach that has not been used previously according to our literature review.

The monitoring system is a cost-effective WIM system consisting of six strain measurement channels and one temperature measurement channel. The testing of the system demonstrated an inconsistent correlation for different cases (the composition of the axle of locomotives and the measuring channel) between the strain signals, speed, and temperature, as well as a strong correlation between the response measured in the strain channels.

Statistical methods were used to demonstrate the quality of our measurements. The sum of medians between the channels of one rail was chosen as the final axle load to compensate for outliers due to dynamic effects. The trend of the heavier first axle in the bogie was identified for the majority of bogies and studied, and the ways to average loads to solve this problem were discussed. The averaging approach *Average bogie* was chosen as a final step to estimate axle loads. The measurement error for the averaged final loads was also presented and determined to be satisfactory for the purposes of this study—e.g., the calibration of the fatigue load model. The mean values and coefficients of variation of the distributions of the axle loads of the most spread locomotives differed from the expected values by no more than 0.3 tonnes and 4.5%, respectively.

Future work should concentrate on three main things. The monitoring system was initially set up to collect traffic load data to study the model used for the estimation of the remaining fatigue life of Norwegian railway bridges in the sense of conservatism. Therefore, the first direction of study should be to use collected measured loads to calculate correction factors for this model and to study how conservative the model based on the most damaging train assumption is. This should include an investigation of the time needed for traffic monitoring to calculate reliable values of correction coefficients—i.e., the study of convergency. A second possible direction of study is investigating whether the accuracy of estimations can be increased and how. A third potential direction for research includes testing the monitoring system at different sites in Norway and comparing the results obtained with the current results.

## Figures and Tables

**Figure 1 sensors-22-01772-f001:**
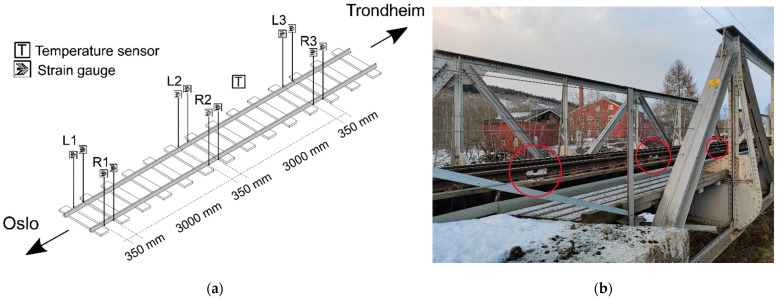
The monitoring system: (**a**) Placement of sensors on the track section; (**b**) Perspective view of the site. The red circles mark the sensors on the rails.

**Figure 2 sensors-22-01772-f002:**
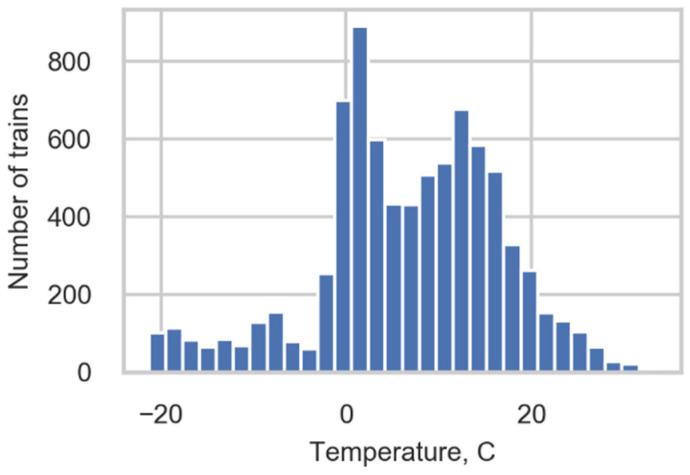
Temperature distribution.

**Figure 3 sensors-22-01772-f003:**
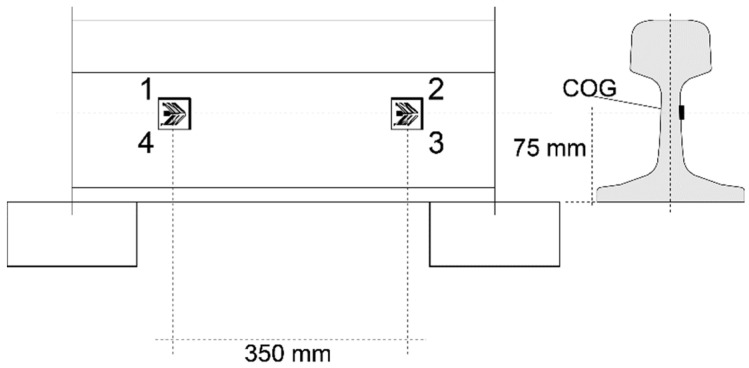
Placement of strain gauges of one channel on the rail cross section.

**Figure 4 sensors-22-01772-f004:**
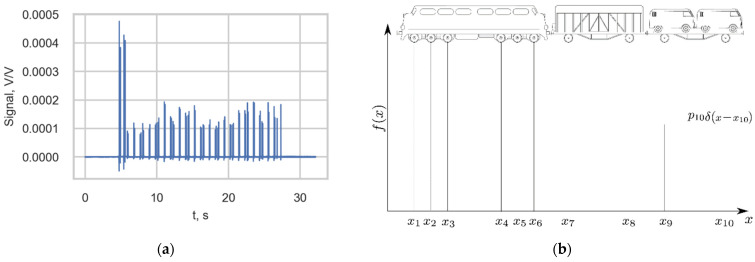
(**a**) Output of one channel of the measurement station triggered by the passage of a passenger train; (**b**) Load function of a train.

**Figure 5 sensors-22-01772-f005:**
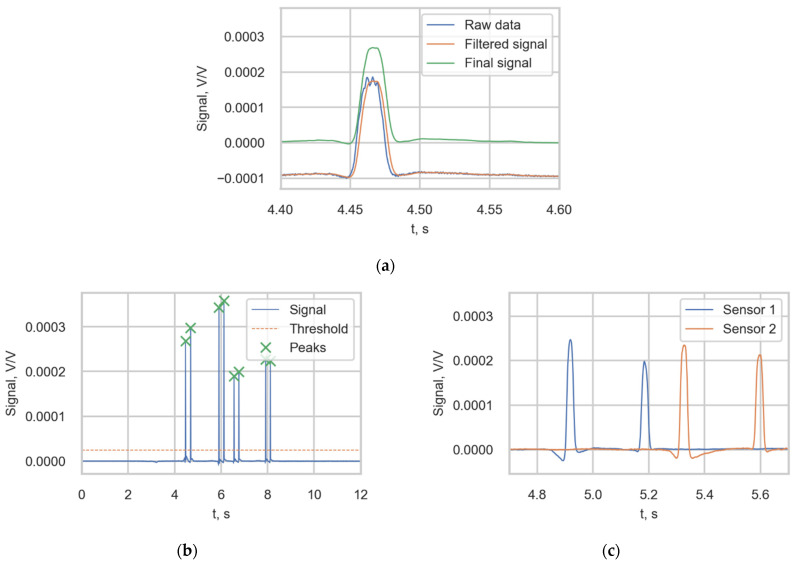
(**a**) Obtaining the final signal from the strain sensor; (**b**) Peak value identification; (**c**) The filtered signals of two adjacent sensors.

**Figure 6 sensors-22-01772-f006:**
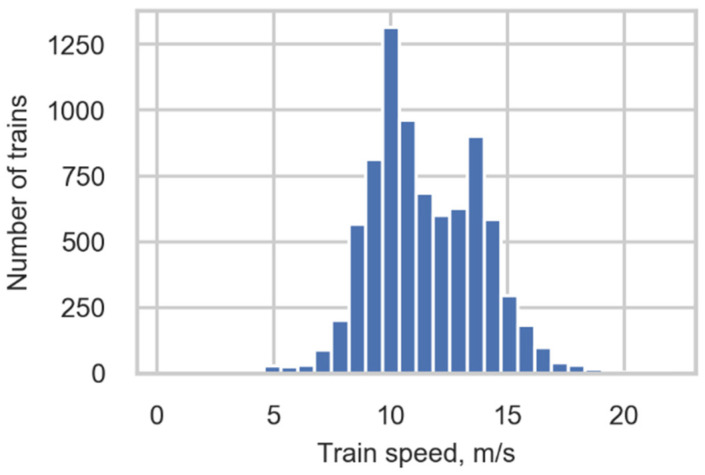
Train speed distribution.

**Figure 7 sensors-22-01772-f007:**
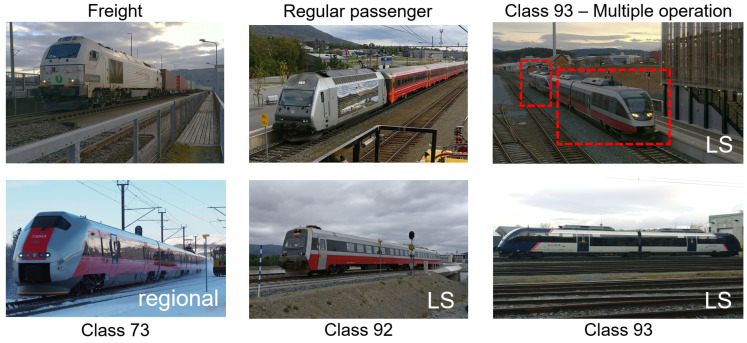
Types of trains in the study. Photograph: NTNU/Stefano Derosa. The red rectangles mark two parts of MU Class 93 in multiple operation case.

**Figure 8 sensors-22-01772-f008:**
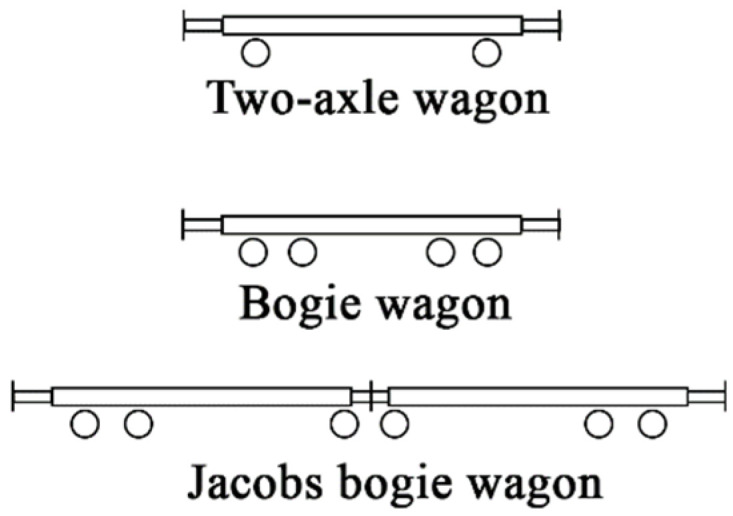
Geometry of two-axle wagon, bogie wagon, and Jacobs bogie wagon.

**Figure 9 sensors-22-01772-f009:**
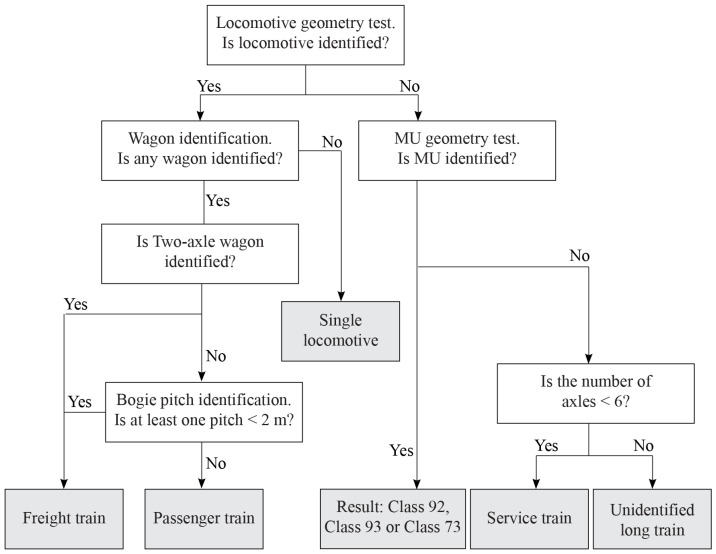
Algorithm used to identify the type of train.

**Figure 10 sensors-22-01772-f010:**
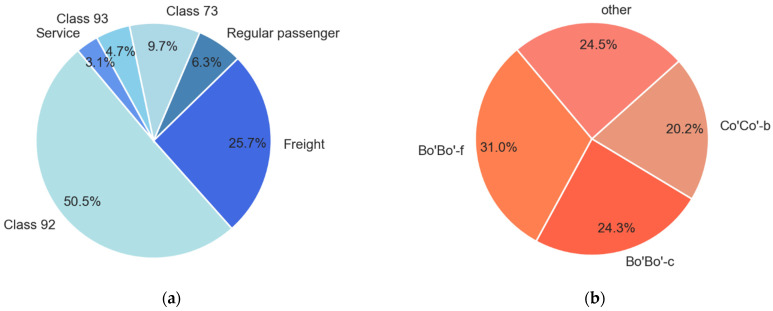
The traffic on the site: (**a**) Train types identified and (**b**) Locomotive types identified.

**Figure 11 sensors-22-01772-f011:**
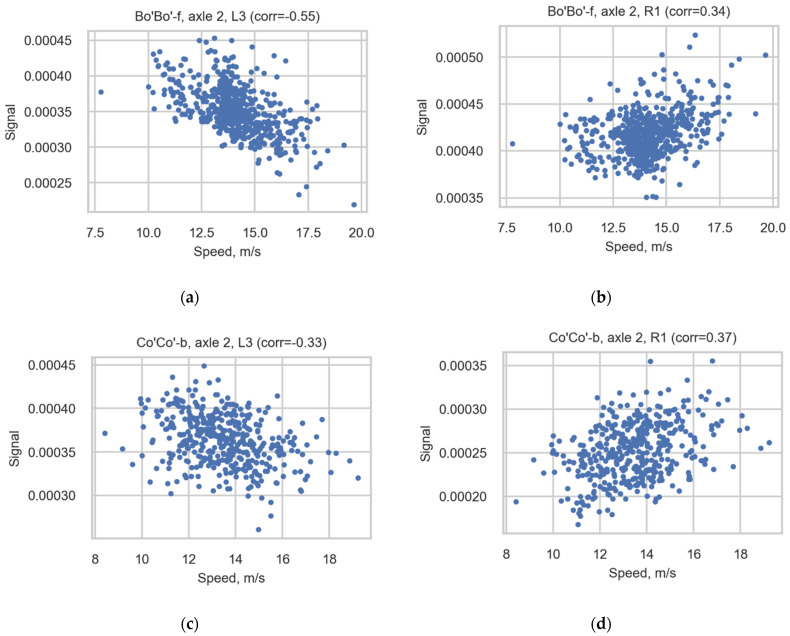
Scatter diagrams illustrating the correlation for axle 2 of Bo’Bo’-f registered in sensors (**a**) L3 and (**b**) R1 and for axle 2 of Co’Co’-b registered in sensors (**c**) L3 and (**d**) R1.

**Figure 12 sensors-22-01772-f012:**
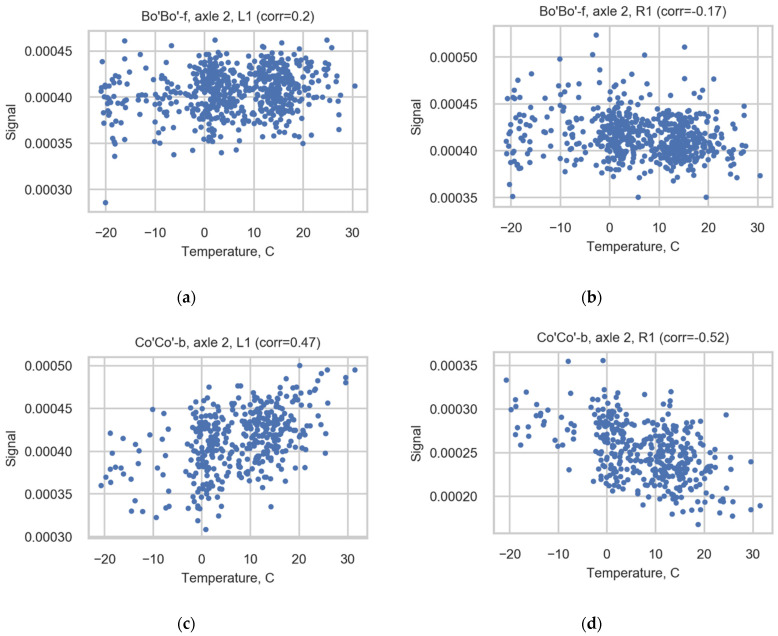
Scatter diagrams illustrating correlation for axle 2 of Bo’Bo’-f registered in sensors (**a**) L1 and (**b**) R1 and for axle 2 of Co’Co’-b registered in sensors (**c**) L1 and (**d**) R1.

**Figure 13 sensors-22-01772-f013:**
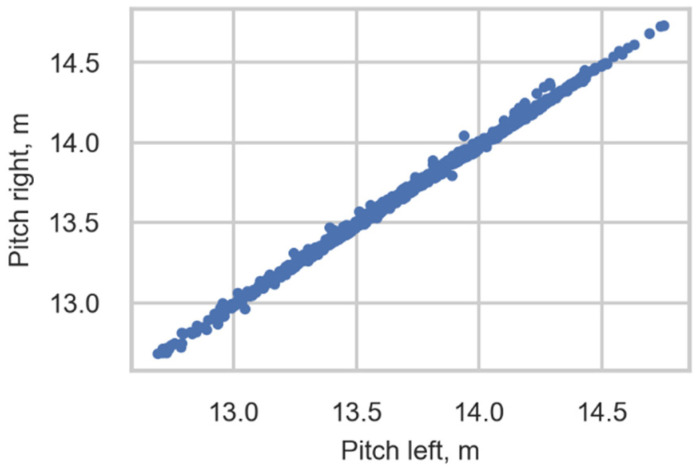
Scatter diagram showing the correlation between pitch obtained from the right rail and left rail. The distance between the second and third axles of Class 92.

**Figure 14 sensors-22-01772-f014:**
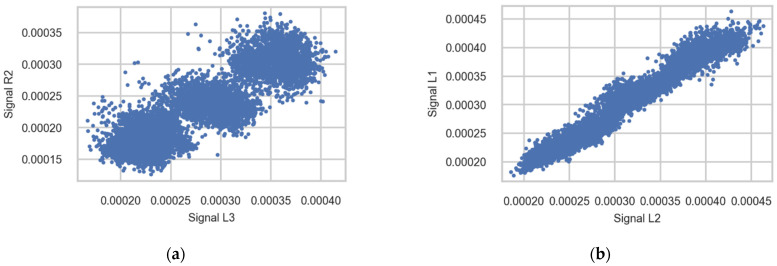
Scatter diagrams showing the correlation between strains measured in different sensors (**a**) R2 and L3 and (**b**) L1 and L2. Signals registered during the passage of the first axle in bogies of Class 92 going to Trondheim.

**Figure 15 sensors-22-01772-f015:**
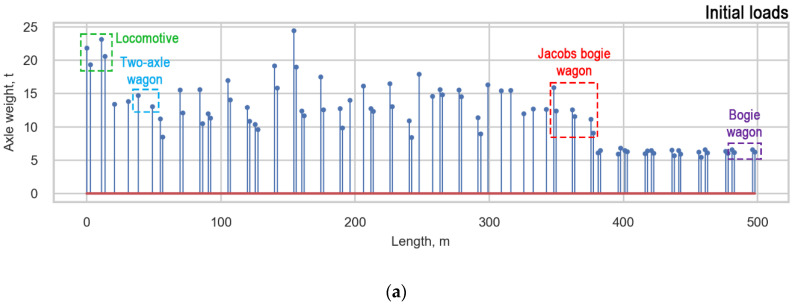
Approaches for averaging obtained loads: (**a**) Initial loads; (**b**) Average bogie; (**c**) Average vehicle, and (**d**) Average design.

**Figure 16 sensors-22-01772-f016:**
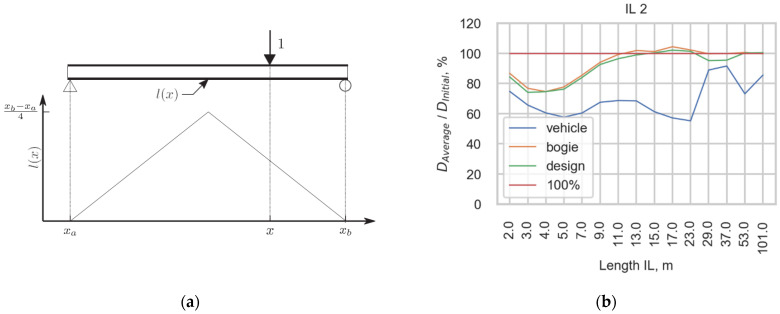
(**a**) One of the most widely spread in the infrastructure IL; (**b**) Comparison of damage calculated for initial load function (100%) and for the three methods for averaging the initial load function, *Average_vehicle*, *Average_bogie*, and *Average_design*.

**Figure 17 sensors-22-01772-f017:**
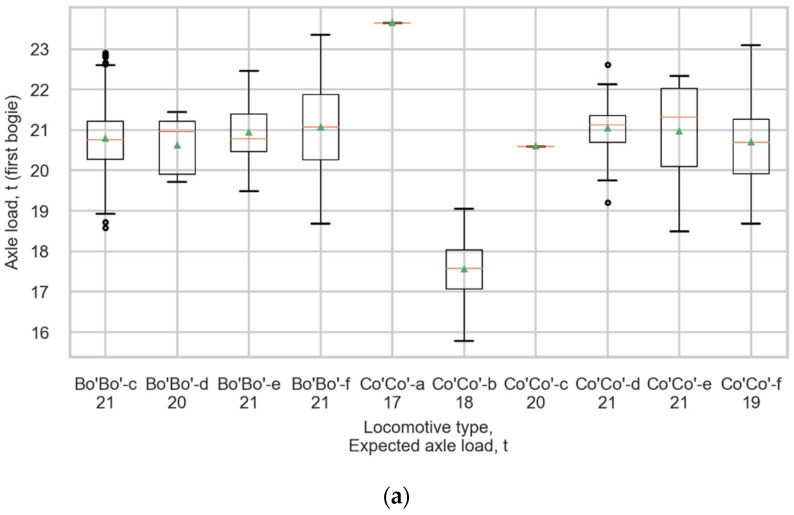
Boxplots representing the distributions of measured loads in all locomotives for (**a**) The first bogie and (**b**) The second bogie.

**Table 1 sensors-22-01772-t001:** The format of files containing (**a**) information about wheels and (**b**) information about trains. Tables are connected through the column PassageDateTime.

**(a)**		
**Key**	**Type**	**Description**
PassageDateTime	DateTime	Date and time when the measurement system detected a train passage.
WheelSpeeds	Float	Estimated speed of wheel when passing the monitoring system.
WheelPitch	Float	Distance from previous wheel. Zero when it is the first wheel of a train passage.
WheelVoltRatL1 WheelVoltRatL2 WheelVoltRatL3 WheelVoltRatR1 WheelVoltRatR2 WheelVoltRatR3	Float	Wheel voltage ratio (signal) estimated by sensors L1-3 and R1-3. See Equation (2) and Section 2.6 for more information.
**(b)**		
**Key**	**Type**	**Description**
PassageDateTime	DateTime	Date and time when the measurement system detected a train passage.
TrainType	String	The train type identified using method described in Section 2.5.
LocomotiveType	String	The locomotive type identified using method described in Section 2.5.
IsElectric	String	The type of the force sets in motion the train; see Section 2.5.
NumberOfAxles	Integer	Number of wheelsets (axles) in the train.
TrainSpeed	Float	Train speed estimated using wheel voltage ratios.
Temperature	Float	Temperature when the measurement system detected a train passage. The output of the temperature sensor is transferred to °C using method described in Section 2.2.
TrainDirection	String	Train running direction. In the current case, there are two options: “to Oslo” and “to Trondheim”.

**Table 2 sensors-22-01772-t002:** Geometries of multiple units.

MU	Number of Axles	Vector of Pitches (m)	Number of Identified Trains
Class 93	6	(2.00, 12.80, 2.80, 12.80, 2.00)	377
Class 92	8	(2.55, 13.90, 2.55, 5.10, 2.55, 13.90, 2.55)	4023
Class 73	16	(2.75, 16.20, 2.75, 4.65, 2.75, 16.20, 2.75, 4.65, 2.75, 16.20, 2.75, 4.65, 2.75, 16.20, 2.75)	774

**Table 3 sensors-22-01772-t003:** Geometries of locomotives.

Locomotive	Number of Axles	Vector of Pitches (m)	Number of Identified Trains
Bo’Bo’-a	4	(3.00, 4.40, 3.00)	0
Bo’Bo’-b	4	(3.20, 4.10, 3.20)	0
Bo’Bo’-c	4	(2.80, 8.20, 2.80)	582
Bo’Bo’-d	4	(2.40, 6.60, 2.40)	9
Bo’Bo’-e	4	(2.70, 5.00, 2.70)	56
Bo’Bo’-f	4	(2.60, 7.80, 2.60)	640
Co’Co’-a	6	(2.00, 2.00, 6.30, 2.00, 2.00)	22
Co’Co’-b	6	(1.80, 1.80, 4.80, 1.80, 1.80)	436
Co’Co’-c	6	(1.80, 2.10, 7.90, 2.10, 1.80)	1
Co’Co’-d	6	(1.80, 1.80, 11.00, 1.80, 1.80)	76
Co’Co’-e	6	(2.00, 2.10, 9.00, 2.10, 2.00)	18
Co’Co’-f	6	(1.80, 2.00, 7.90, 2.00, 1.80)	50

**Table 4 sensors-22-01772-t004:** Identification of whether a train is electric.

IsElectric	MU	Locomotive
True	Class 73	Co’Co’-b, Bo’Bo’-a, Bo’Bo’-b, Bo’Bo’-c, Bo’Bo’-e, Bo’Bo’-f
False	Class 93, Class 92	Co’Co’-a, Co’Co’-c, Co’Co’-d, Co’Co’-e, Co’Co’-f, Bo’Bo’-d

**Table 5 sensors-22-01772-t005:** Reference axle and wheel loads of locomotives and their cumulative weights.

Locomotive Type	Axle Load, t	Wheel Load, t	Locomotive Weight, t
Bo’Bo’-c	21	10.5	84
Bo’Bo’-d	21	10.5	84
Bo’Bo’-e	20	10	80
Bo’Bo’-f	21	10.5	84
Co’Co’-a	17	8.5	102
Co’Co’-b	18	9	108
Co’Co’-c	20	10	120
Co’Co’-d	21	10.5	126
Co’Co’-e	21	10.5	126
Co’Co’-f	19	9.5	114

**Table 6 sensors-22-01772-t006:** (**a**) Mean value and (**b**) Coefficient of variation for wheel loads calculated for channels and axle load of Bo’Bo’-f. The expected axle load is 21 tonnes and the wheel load is 10.5 tonnes.

**(a) MEAN**							
**Level**	**L1, t**	**L2, t**	**L3, t**	**R1, t**	**R2, t**	**R3, t**	**Axle Load, t**
axle 1	10.6	11.3	10.1	12.1	12.1	11.2	22.8
axle 2	9.7	9.7	9.5	10.1	9.9	10.1	19.7
axle 3	11.2	12.0	11.0	11.4	11.3	10.4	22.6
axle 4	9.9	9.5	9.2	9.9	9.9	10.6	19.7
**(b) COV**							
**Level**	**L1, %**	**L2, %**	**L3, %**	**R1, %**	**R2, %**	**R3, %**	**Axle Load, %**
axle 1	10.9	10.2	9.9	10.9	11.1	8.0	8.7
axle 2	5.9	7.5	9.2	5.6	8.9	6.3	3.0
axle 3	10.1	10.1	8.6	8.6	10.0	8.8	7.9
axle 4	6.5	6.0	11.4	6.4	6.5	6.5	3.8

**Table 7 sensors-22-01772-t007:** Comparison of expected weights of locomotives and mean values of measured weights of all locomotives according to their type. Coefficients of variation are also presented.

Locomotive	Number of Trains	Expected Weight, t	Measured Weight (Mean, t; cov, %)
Bo’Bo’-c	53	84	84.1, 2.9%
Bo’Bo’-d	198	84	82.3, 3.5%
Bo’Bo’-e	76	80	85.4, 3.5%
Bo’Bo’-f	721	84	84.7, 4.1%
Co’Co’-a	9	102	142.5, 0.0%
Co’Co’-b	484	108	107.8, 3.2%
Co’Co’-c	1	120	122.5, 0.0%
Co’Co’-d	65	126	126.7, 3.3%
Co’Co’-e	25	126	124.8, 4.9%
Co’Co’-f	46	114	124.3, 4.9%

**Table 8 sensors-22-01772-t008:** Comparison of expected loads and mean values accompanied by the coefficient of variation for axle loads.

**Locomotive** **(Number)**	**Expected Load, t**	**Axle 1**	**Axle 2**	**Axle 3**	**Axle 4**	**Locomotive** **Axles, Mean**
Bo’Bo’-c(532)	21	21.6, 6.8%	20.3, 2.7%	21.9, 5.9%	20.2, 3.0%	21.0, 6.2%
Bo’Bo’-d (9)	21	21.7, 4.3%	19.4, 3.5%	22.0, 5.2%	19.3, 4.0%	20.6, 7.4%
Bo’Bo’-e (53)	20	22.1, 6.5%	20.0, 2.2%	22.8, 7.6%	20.4, 3.2%	21.3, 7.9%
Bo’Bo’-f(623)	21	22.8, 8.7%	19.7, 3.0%	22.6, 7.9%	19.7, 3.8%	21.2, 9.7%
**Locomotive** **(Number)**	**Expected Load, t**	**Axle 1**	**Axle 2**	**Axle 4**	**Axle 5**	**Locomotive** **Axles, Mean**
Co’Co’-a (1)	17	28.0, 0.0%	22.2, 0.0%	26.8, 0.0%	22.9, 0.0%	23.7, 11.3%
Co’Co’-b(410)	18	20.0, 8.7%	16.4, 4.1%	21.1, 8.3%	16.7, 4.8%	18.0, 12.3%
Co’Co’-c (1)	20	23.1, 0.0%	20.2, 0.0%	21.8, 0.0%	19.7, 0.0%	20.4, 7.7%
Co’Co’-d (41)	21	23.5, 6.6%	21.3, 3.2%	22.6, 7.4%	21.0, 4.4%	21.1, 9.4%
Co’Co’-e (18)	21	22.1, 7.6%	22.8, 7.4%	21.5, 5.8%	21.2, 5.2%	20.8, 10.1%
Co’Co’-f (43)	19	23.0, 6.2%	21.5, 6.3%	22.2, 6.8%	21.4, 7.5%	20.7, 11.3%

**Table 9 sensors-22-01772-t009:** Final measurement error of the monitoring system.

**Locomotive** **(Number)**	**Expected Load, t**	**Axle 1**	**Axle 2**	**Axle 3**	**Axle 4**	**Expected Weight, t**	**Measured** **Weight, t**
Bo’Bo’-c (532)	21	21.0, 3.5%	21.0, 3.5%	21.1, 3.1%	21.1, 3.1%	84	84.1, 2.9%
Bo’Bo’-d (9)	21	20.7, 3.2%	20.7, 3.2%	20.9, 3.3%	20.9, 3.3%	84	83.3, 2.9%
Bo’Bo’-e (53)	20	21.1, 3.5%	21.1, 3.5%	21.6, 4.2%	21.6, 4.2%	80	85.4, 3.5%
Bo’Bo’-f (623)	21	21.2, 4.5%	21.2, 4.5%	21.1, 4.2%	21.1, 4.2%	84	84.7, 4.1%
**Locomotive** **(Number)**	**Expected Load, t**	**Axle 1**	**Axle 2**	**Axle 4**	**Axle 5**	**Expected Weight, t**	**Measured** **Weight, t**
Co’Co’-a (1)	17	24.0, 0.0%	24.0, 0.0%	23.5, 0.0%	23.5, 0.0%	102	142.5, 0.0%
Co’Co’-b (410)	18	17.7, 3.6%	17.7, 3.6%	18.3, 3.8%	18.3, 3.8%	108	107.8, 3.2%
Co’Co’-c (1)	20	20.7, 0.0%	20.7, 0.0%	20.1, 0.0%	20.1, 0.0%	120	122.5, 0.0%
Co’Co’-d (41)	21	21.2, 3.3%	21.2, 3.3%	21.0, 3.8%	21.0, 3.8%	126	126.7, 3.3%
Co’Co’-e (18)	21	21.1, 5.7%	21.1, 5.7%	20.5, 4.2%	20.5, 4.2%	126	124.8, 4.9%
Co’Co’-f (43)	19	20.8, 5.0%	20.8, 5.0%	20.7, 4.9%	20.7, 4.9%	114	124.7, 4.5%

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
