# Peer review of "Train Classification Using a Weigh-in-Motion System and Associated Algorithms to Determine Fatigue Loads"

_sensors, 2022, doi:10.3390/s22051772_

Round 1

Reviewer 1 Report

The paper presents a study about a WIM to estimate axle loads and distinguish different train types based on the axle distribution. This is an interesting subject that deserves to be studied. However, before publication, the following comments should be considered and answered to improve the quality of the work.

1 – Abstract/Title/Introduction: The paper focus on WIM system to classify different train types and estimate axle loads. Then, averaging is done to get possible load models for fatigue analysis. However, 90% of the paper is related with the identification of axle loads and type of trains, with just a slight introduction to fatigue. The reviewer thinks that the importance given to fatigue in the introduction (4 paragraphs) is disproportional. On the other hand, no reference to fatigue is given in the abstract, but it appears again in the title. The importance of each topic in the paper should be more distinguished, emphasizing in the introduction the main novelties of this work.

2 – Section 2.2: Some information regarding this specific parameters A, B and C should be given. Are they typical values adopted for these circumstances?

3 – Fig.3: Each strain gauge has 2 measuring grids in parallel? Because there are only 2 strain gauges per rail in each channel, correct? Moreover, it is stated that “the longitudinal distance between each sensor is nominally 3 m”, but in fact is the distance between each pair of sensors, correct?

4 – Section 2.5: The assumption of equally distributed load between wheels is assumed in this work. However, this assumption is not always valid, especially in freight trains with unbalanced loads. Did the authors think about a different way to obtain the calibration coefficient without considering this assumption?

5 – Section 3.2: Which correlation coefficient was used in this analysis presented in section 3.2?

6 – Table 11: The authors refer that “The possible explanation of the effect (higher load of the first axle of each bogie) is that the first axle of bogie is subjected to dynamic effects.”. What do the authors mean with “subjected to dynamic effects”? Aren’t all axles subjected to dynamic effects?

7 – Fig. 16: It is not clear how the damage was computed: which bridge is being analyzed here? Which S-N curves and which details were analyzed?

8 – Conclusions: conclusions should avoid references and citations.

9 – General: The paper has too many tables. Consider reduce the amount of information in the tables or send some of them to appendices.

Author Response

We take this opportunity to thank the reviewer for spending his/her precious time to provide insightful and constructive feedback on our work. We are indebted to the reviewer for contributing to better presentation and clarity of our manuscript. We welcome any other comments the reviewer may have.

Please see the attachment for Response report. The document consists of four sections:

  1. Response to comments from reviewers
  2. Tracking of changes
  3. Language editing
  4. Revised manuscript

Reviewer 2 Report

Weigh-in-motion (WIM) has many applications, it is not clear why the Authors focused title and introduction on fatigue only. In this way they failed to properly describe the contents of their submission. On the other hand, the abstract clearly summarizes the actual contents and the word “fatigue” is never mentioned.

It is opinion of this Reviewer that the Authors should focus their attention to clearly highlight what is new in their contribution. Different WIM methodologies already exist, see for examples the following journal articles that should be quoted and commented:   

- Development of a dynamical weigh in motion system for railway applications, Meccanica 2016, 51(10), pp. 2509-2533

- Weigh-in-motion of train loads based on measurements of rail strains, Structural Control and Health Monitoring 2021, 28(11), e2818

- Vertical wheel-rail force waveform identification using wavenumber domain method. Mechanical Systems and Signal Processing 2021, 159, 107784

Please, check for more article on WIM procedures in railways and significantly shorten (or even remove) the long discussion about fatigue, eliminating the relevant references. The title should be changed into something that clearly communicates the actual contents, e.g., Strain-based weigh-in-motion procedure for railways: validation for a Norwegian steel bridge (I’m sure that the Authors can come up with a much better title).

Author Response

(The authors gave the same response as above.)

Reviewer 3 Report

Some detail comments are as follows,

1) The authors have published four articles in the same topic. The new developments of the paper should be justified. Authors need clearly justify the novelty of the paper, especially comparing with previous publications. 

2) The long sentence should be replaced with short sentences, such as "The wheel pitches..." is not clear.

3) In Equation (3), what is "dt'"?

4) In Equation (4), who does it mean "mean(t2-t1)"? what is "dt1"?

5) The English writing needs to be improved. for example

Line 190, "...between ..." should be "...between the peaks of measurement from two adjacent sensors as shown in Equation (3)".

Author Response

(The authors gave the same response as above.)

Reviewer 4 Report

This study focus on the detection of actual traffic conditions, which is essential in assisting evaluation of structural performance.

And this paper was written in a good organization, I can clear understand the main point and intereted details simultaneously. However, more conclusive sentences could be better.

Another advice is that weighs of goods and fuels may be considered when we detect axle loads, then the calibration coefficients are different according to specific axles.

Author Response

(The authors gave the same response as above.)

Round 2

Reviewer 1 Report

My comments were properly acknowledged. The paper can be accepted in its current form.

Author Response

thank you for your review

Reviewer 2 Report

The comments of this Reviewer have been properly addressed. No further concerns.

Author Response

thank you for your review